# Microstructures and Properties of Auto-Tempering Ultra-High Strength Automotive Steel under Different Thermal-Processing Conditions

Han Jiang [1], Yanlin He [1,*], Li Lin [2], Rendong Liu [2,3], Yu Zhang [3], Weisen Zheng [1] and Lin Li [1]

[1] School of Materials Science and Engineering, Shanghai University, Shanghai 200044, China; 18362208055@163.com (H.J.); wszheng@shu.edu.cn (W.Z.); liling@shu.edu.cn (L.L.)
[2] Ansteel Group Co., Anshan 114021, China; ag_linli@126.com (L.L.); ag_lrd@126.com (R.L.)
[3] Ansteel Group Beijing Research Institute, Beijing 102211, China; 15104124606@163.com
[*] Correspondence: ylhe@t.shu.edu.cn

**Abstract:** Automotive steels with ultra-high strength and low alloy content under different heating and cooling processes were investigated. It was shown that those processes exhibited a great influence on the performance of the investigated steels due to the different auto-tempering effects. Compared with the steels under water quenching, there was approximately a 70% increase in the strength and elongation of steels under air cooling, in which the martensite was well-tempered. Although the elongation of the steel with a microstructure composed of ferrite, well-tempered martensite and less-tempered martensite could exceed 15%, the hole expansion ratio was still lower because of the undesirable hardness distribution between the hard phases and the soft phases. It followed from the calculation results based on SEM, TEM and XRD analyses, that for the steel under air cooling, the strengthening mechanism was dominated by the solid solution strengthening and the elongation was determined by the auto-tempering of martensite. Experiments and analyses aimed to explore the strengthening and plasticity mechanisms of auto-tempering steels under the special process of flash heating.

**Keywords:** ultra-high strength steel; auto-tempering; martensite; hole expansion ratio; flash heating





## 1. Introduction

With the increasing demand for energy saving and environmental protections, the strength level of the automotive steel sheet has been raised and many works have contributed to the development of steels that perform better [1–3]. These steels are referred to as Advanced High Strength Steel (AHSS), among which, twinning-induced plasticity (TWIP) steels are characterized by an austenite phase with extremely large uniform elongation and high ultimate tensile strength [4,5]. However, the high Mn content (15–30 wt%) in the steels leads to high costs as well as problems in the production line which limits its further application in car industries. In addition, quenching-partition (Q&P) steel is designed to be quenched into a certain temperature range between $M_S$ and $M_f$, then carbon partition from the martensite to the retained austenite starts, which results in a certain amount of austenite that is stabilized at room temperature to ensure the excellent strength and plasticity of the steel [1]. Though there are no expensive elements included, its complex process route does not fit the present continuous annealing line (CAL) and supplemental investment is required. In recent years, in view of the urgent demand for green manufacturing in the iron and steel industries and for advanced manufacturing processes to produce steel with a high performance, the near net shape process represented by compact strip production (CSP) to produce automotive steel has been widely studied [6,7]. Obviously, steel with a high alloy content and a complex process is not an ideal candidate for this special process.

As is well known, auto-tempering is a phenomenon in which the first-formed martensite near the martensitic transformation start temperature ($M_S$) is tempered during the following up process of quenching [8]. When auto-tempering happens, martensite in steel with a lower carbon content can be decomposed and the mechanical performance is improved due to the formation of tempered martensite and metastable carbides. Thus, the steel exhibits better strength and plasticity. Recently, some works have developed auto-tempering steel with a strength above 1200 MPa and a good elongation above 10% [9–11], but the strengthening and plasticity mechanisms of the steel under different thermal-processing conditions are still obscure. In the present work, the effect of auto-tempering on microstructures and the properties of two low alloy steels with different carbon contents are investigated to elucidate the intrinsic mechanism.

## 2. Experimental Procedure

The chemical compositions of the steel used in the present work are listed in Table 1. Ingots were prepared by pure raw materials and vacuum induction melting at $10^{-1}$ Pa vacuum value. Slabs with a 35 mm thickness were hot rolled after reheating at 1200 °C to produce a 3.5 mm thick sheet. The hot-rolled sheets were pickled and cold rolled to a 60 pct reduction. The samples with a size of $\varphi 4 \times 10$ mm heated to 880 °C at a rate of 2 k/s, were prepared to measure the phase transformation temperature with a DIL805 thermal expansion analyzer, and liquid nitrogen was used in the quenching process. The dilatometric curves of the samples are shown in Figure 1 and the experimental results are listed in Table 1.

**Table 1.** Chemical compositions (wt%) and the phase transformation temperature (°C) of the investigated steels.

| Steel | C | Mn | Si | Cr | Ti | $A_{C1}$ | $A_{r3}$ | $M_S$ |
|-------|------|------|------|------|-------|----------|----------|-------|
| A | 0.13 | 2.10 | 1.35 | 0.98 | 0.010 | 728 | 819 | 395 |
| B | 0.18 | 2.13 | 1.40 | 1.00 | 0.012 | 715 | 833 | 371 |

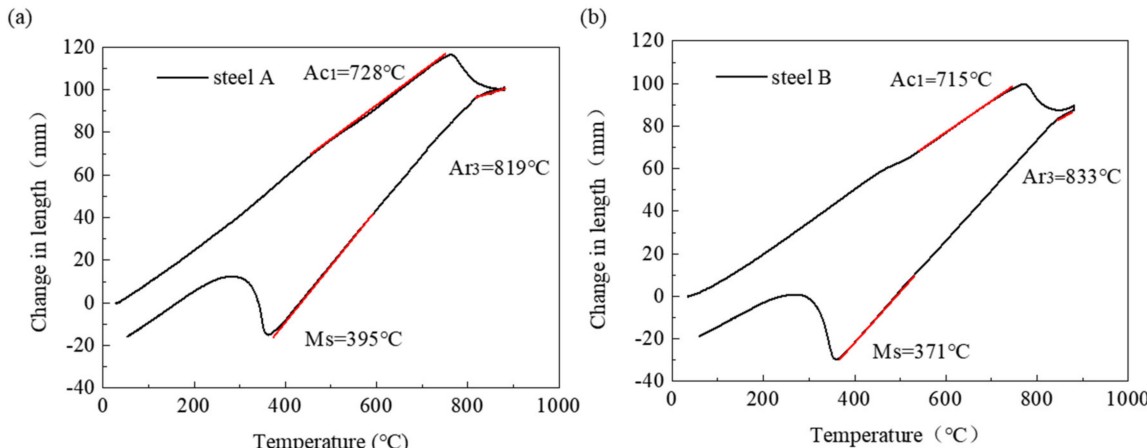

**Figure 1.** Dilatometric curves and phase transformation temperature of the experimental steels: (**a**) steel A and (**b**) steel B.

Steels A and B with different compositions were designed to compare the effect of carbon on the auto-tempering behavior. The heat-treatment cycle is schematically shown in Figure 2, and was operated in salt bath furnace. The samples were austenized at 880 °C and intercritical annealed at 800 °C for 3 min and then air cooled to room temperature to compare the auto-temper behavior of the full martensite microstructure and the ferrite–martensite microstructure. Three cooling processes, such as air cooling (denoted by the red line), water quenching (denoted by the blue line) and quenching to 200 °C in a salt bath for 3 and 30 min and then air cooled to room temperature (denoted by the green line)

were applied to study the effect of different cooling procedures on auto-temper behavior. In addition, by using Gleeble3500 (DSI, Saint Paul, MN, USA), flash-heating with a heating rate 300 °C/S was adopted to study the effect of heating processes on auto-temper behavior.

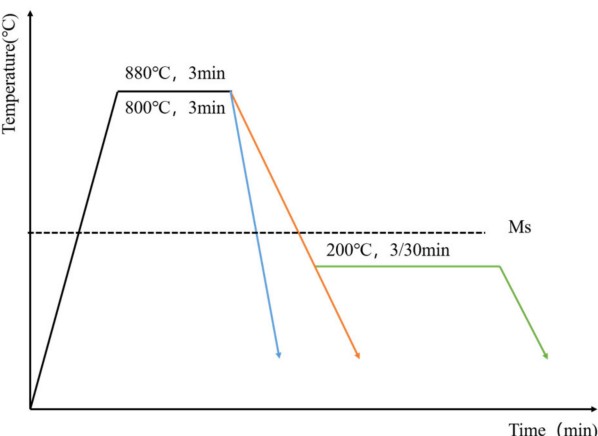

**Figure 2.** Schematic diagram of heat treatments.

The tensile properties of the samples were measured by the uniaxial tensile test with the standard specimen (according to the GB/T228.1-2010 standard, gauge length: 50 mm, width: 12.5 mm, thickness: 1.2 mm), the tensile direction of which was paralleled to the rolling direction. The tests were performed by a MTS C45.305E electronmechanical universal testing machine with a strain rate of about $1 \times 10^{-3}$ S$^{-1}$ at room temperature, and an extensometer was used to measure the strain of the gauge length of the samples. The experimental results were determined by the average value of three tensible samples. The microstructures of the samples were etched with picric acid and 4% of nital was observed by means of OM and SEM analysis, respectively. Prior austenite grain size was measured by the linear intercept method (according to GB/T 3488.2 standard). The samples for transmission electron microscopy (TEM) were sliced from bulk specimens and mechanically polished to thick discs of about 50 μm with a diameter of 3 mm. Electrolytic polishing was conducted using 10 vol.% perchloric acid in ethanol at −35 °C in a twin-jet electrolytic polisher. The samples prepared were detected in a JEM-2010F microscope (JEOL Ltd, Tokyo, Japan) with an accelerated voltage of 200 kV. To calculate the dislocation density, specimens were measured in a 18KW D/MAX2500 X-ray diffractometer (Rigaku, Tokyo, Japan) with Cu-Kα radiation. Scanning was carried out with a 0.02° step and a 3 s stay for each step over a 2θ range from 40° to 100°.

The hole expansion test (HET) was carried out according to the ISO16630-2009 standard using ITC-SP225 equipment with a sample size of 90 mm × 90 mm × 1.2 mm and the diameter of the initial hole $D_0$ was 10 mm in the sample center. The test speed was 3 mm/min and the sample blank holder force was 50 KN. The hole expansion ratio (HER) was calculated by the following equation [8]:

$$\text{HER}(\%) = (D_f - D_0)/D_0 \times 100\% \tag{1}$$

where $D_0$ and $D_f$ represent the initial hole diameter (mm) and the ultimate hole diameter (mm) when a crack is initiated, respectively.

## 3. Results and Discussion

### 3.1. Effects of Various Cooling Processes on the Microstructures and the Mechanical Properties of the Experimental Steels

Figure 3 shows the microstructures of steel A and steel B heated at 880 °C for 3 min after being treated by water-quenching, and after being air-cooled and soaked at 200 °C for 3 min during the air-cooling. Combined with Figure 4, it can be seen that the microstructures of

the experimental steels are mainly composed of lath martensite and film retained austenite. Compared with the structures after water-quenching, the lath martensite in the steels following air-cooling looks broad and the boundary is blurry, and tempered martensite (TM) is formed. In addition, the tempering of the martensite in steel A with the lower carbon content is more obvious than that in steel B. As shown in Figure 5, nano-scale TiC carbide can be observed in the experimental steel.

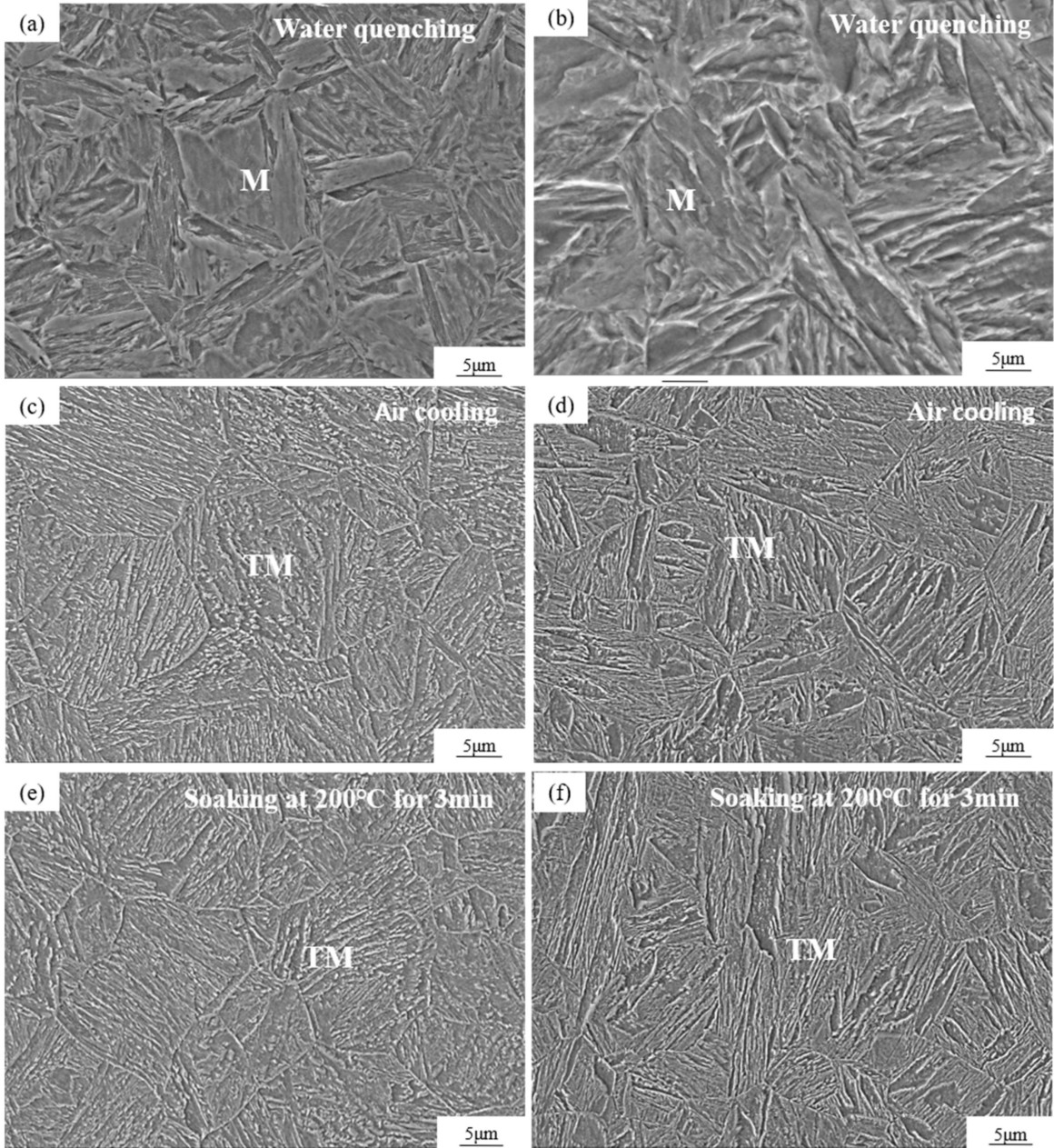

**Figure 3.** Microstructures of steel A (**a,c,e**) and steel B (**b,d,f**) heated at 880 °C for 3 min then treated with different cooling processes.

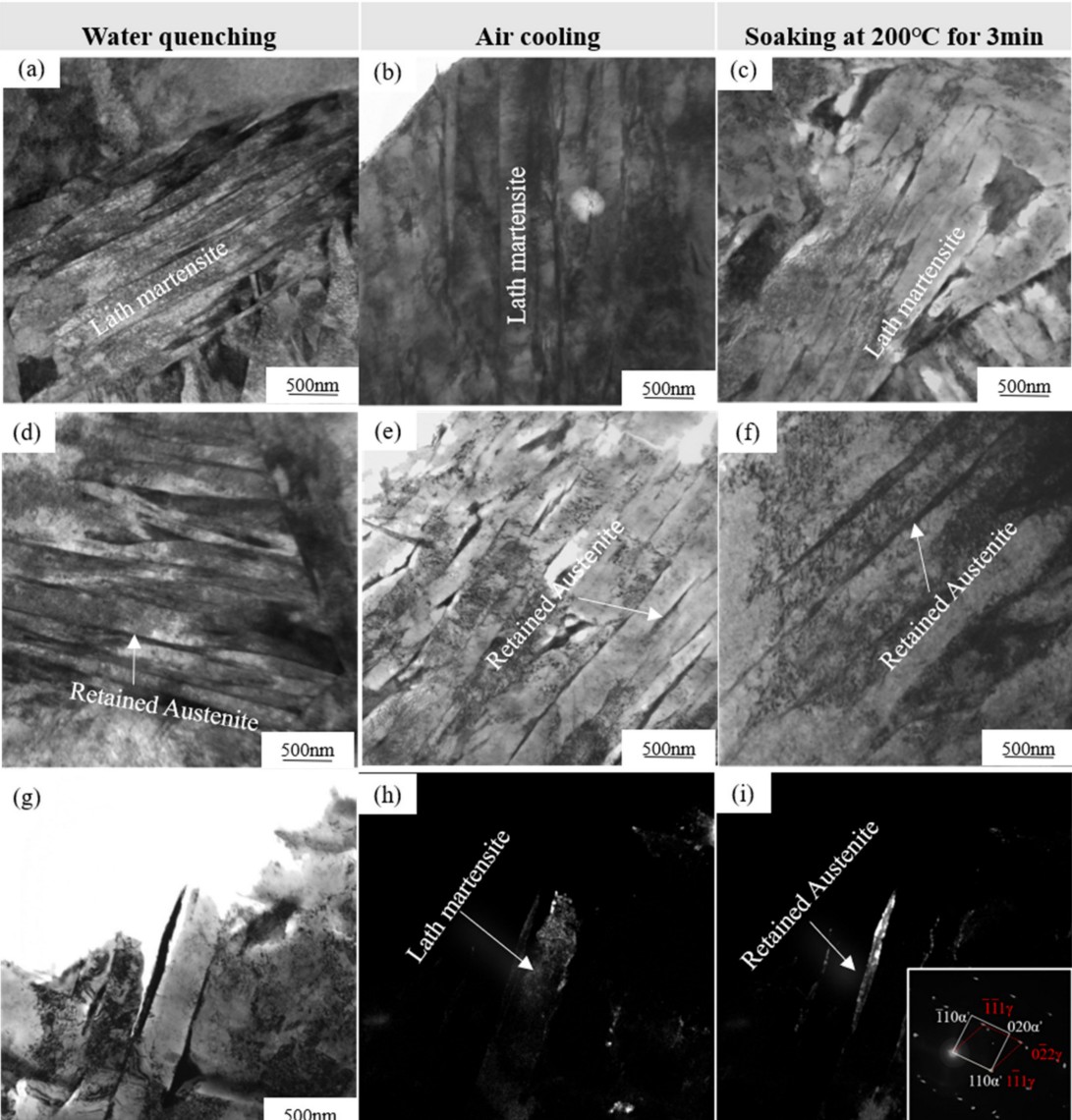

**Figure 4.** TEM analysis of lath martensite and retained austenite in steel A (**a**–**c**) and steel B (**d**–**f**) heated at 880 °C for 3 min then treated by different cooling processes, (**g**–**i**) show the bright field image, the dark field image and the corresponding selected area electron diffraction (SAED) pattern.

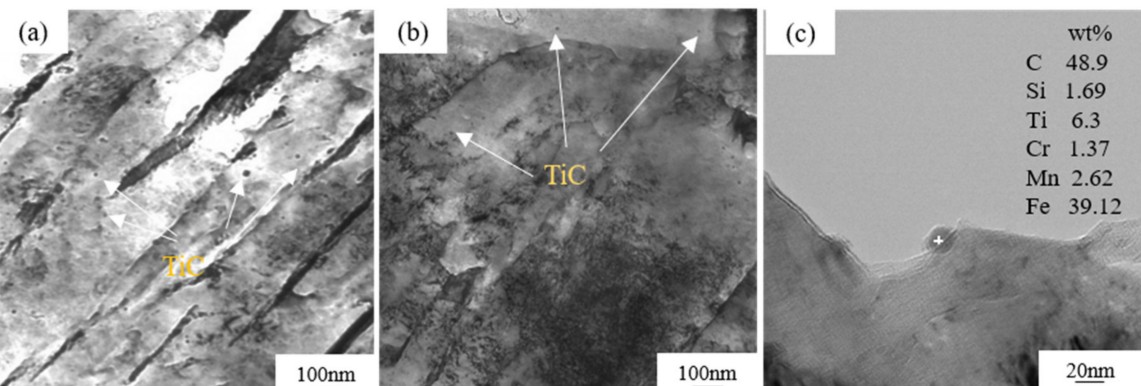

**Figure 5.** TEM analysis of nano-scale TiC carbides precipitated in experimental steels heated at 880 °C for 3 min then air cooling: (**a**) steel A, (**b**) steel B and (**c**) chemical composition of TiC.

Figure 6 shows the microstructures of steel A and steel B soaked at 200 °C for 30 min during air cooling. Compared with those soaked at 200 °C for 3 min, the martensite lath of the experimental steels is obviously coarsened. In steel B, the precipitation of ε-carbides can also be found, as shown in Figure 7. So, the yield strength is further improved by the interactions between nano-scale carbides and dislocations.

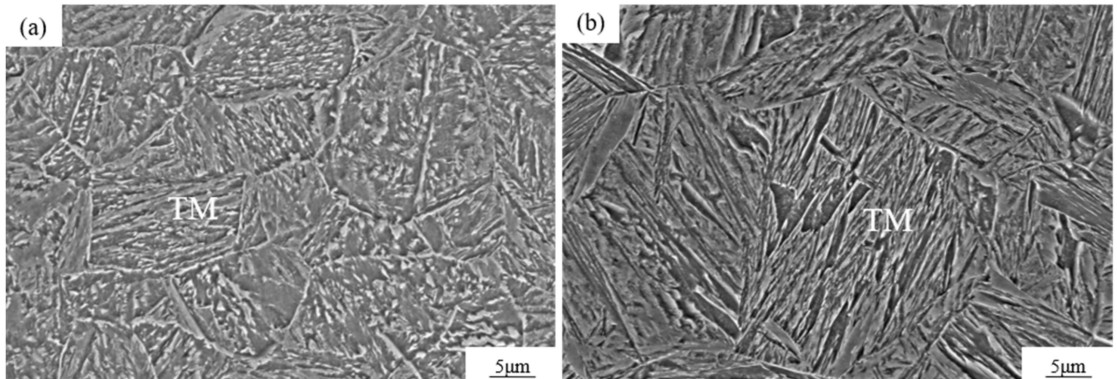

**Figure 6.** Microstructures of steel A (**a**) and steel B (**b**) heated at 880 °C for 3min then soaked at 200 °C for 30 min during air cooling.

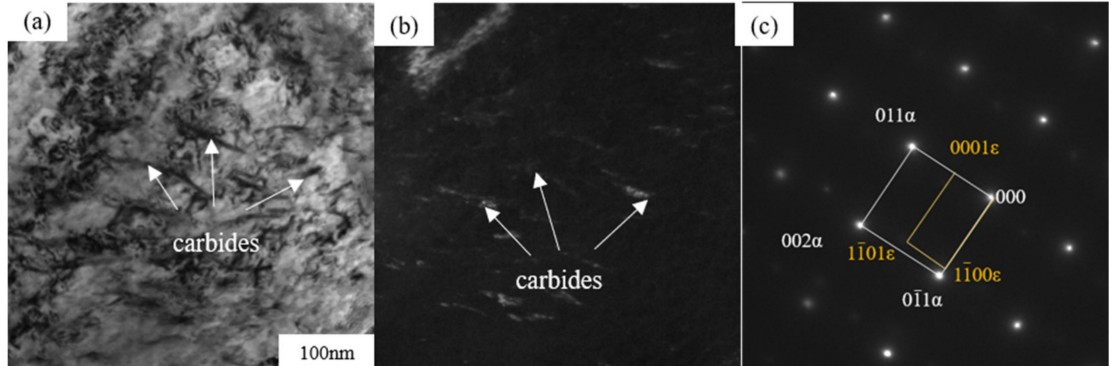

**Figure 7.** TEM analysis of steel B heated at 880 °C for 3 min, soaked at 200 °C for 30 min then air-cooled. (**a**) bright field TEM image and (**b**) dark field TEM image and (**c**) corresponding SAED pattern of lath martensite and ε-carbides.

The tensile properties at room temperature of the experimental steels under different cooling processes are shown in Table 2. The tensile strength of all the steels exceeded 1000 MPa. The steels which underwent water quenching had the highest tensile strength and the lowest elongation. For steel A, the product of strength and elongation (PSE) under the air-cooling condition was the highest, while for steel B, the highest PSE was obtained under soaking at 200 °C for 3 min during air cooling.

**Table 2.** Mechanical properties of steels heated at 880 °C for 3 min then treated by different cooling processes.

| Cooling Process | TS (MPa) | | YS (MPa) | | TE (%) | | PSE (MPa·%) | |
|---|---|---|---|---|---|---|---|---|
| | A | B | A | B | A | B | A | B |
| Water quenching | 1384 ± 30 | 1646 ± 39 | 1184 ± 33 | 1242 ± 37 | 5.9 ± 0.2 | 6.6 ± 0.3 | 8166 | 10,864 |
| Air cooling | 1258 ± 35 | 1466 ± 23 | 843 ± 29 | 916 ± 26 | 10.9 ± 0.4 | 8.9 ± 0.2 | 13,712 | 13,047 |
| Soaking at 200 °C for 3 min | 1195 ± 13 | 1410 ± 24 | 752 ± 21 | 826 ± 17 | 11.4 ± 0.7 | 10.1 ± 0.5 | 13,623 | 14,241 |
| Soaking at 200 °C for 30 min | 1176 ± 22 | 1422 ± 31 | 743 ± 14 | 1038 ± 13 | 8.1 ± 0.3 | 9.0 ± 0.1 | 9526 | 12,798 |

As shown in Table 2, compared with the samples under water quenching, there was approximately a 70% increase in the PSE of steel A under air cooling, while there was

a 20% increase for steel B under the same cooling conditions. As indicated in Table 1, the martensite transformation temperature $M_S$ for steel A and steel B were 395 °C and 371 °C, respectively. When martensite is formed below $M_S$, it may have the opportunity of tempering during the remainder of the cooling. This phenomenon, which is referred to as auto-tempering, is more likely to occur in steels with a higher $M_S$, when the temperature is about or above 300 °C. This is because at this temperature, carbon possesses a diffusion activation energy between 60 and 80 kJ mol$^{-1}$, which is favorable to the diffusion in martensite, i.e., the interstitial carbon atoms in the tetragonal martensite lattice can easily diffuse from the octahedral interstices to the position of defects such as dislocations and/or the martensite boundary [12,13]. Then the solid solution strengthening effect of martensite is weakened, and the elongation increases as the tensile strength decreases. Compared with steel B, the $M_S$ of steel A with a lower carbon level is higher, so auto-tempering is more likely to take place.

Detected by an infrared thermometer, the average cooling rates during the martensite transformation (between 400 and 20 °C) for the samples were about 300 °C/S and 3 °C/S, respectively. The mean diffusion distance of C atoms $d_c$ in martensite can be integrated by Equations (2) and (3) [14–16]:

$$d_c = \sqrt{Dt} \tag{2}$$

$$D = 2 \times 10^{-6} \times \exp(-1.092 \times 10^5 / 8.314T) \tag{3}$$

where $t$ is the time (s), $D$ is the diffusion coefficient of carbon and $T$ is the temperature (K). The diffusion distances of the C atoms corresponding to water quenching and air cooling are plotted in Figure 8. Carbon migrated 3.8 μm in the first martensite laths of steel A under air cooling to room temperature while under water quenching, the value was only 0.95 μm. Similarly, for steel B, the diffusion distances were 2.5 μm and 0.73 μm, respectively. According to the calculated results, it can be inferred that for the steel under water quenching, the probability of the occurrence of auto-tempering is less because of the limited diffusion capacity of the carbon atom.

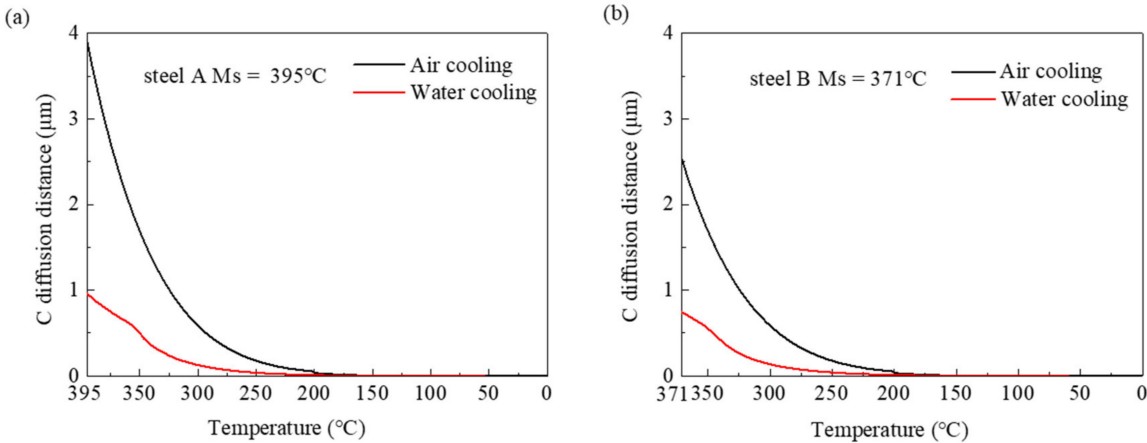

Figure 8. C diffusion distance during martensite transformation for steel A (a) and steel B (b).

In order to have a clear understanding of the strengthening mechanisms of the two steels under air cooling, all strengthening factors which play a role in $\sigma_Y$ were carefully considered. According to the previous study [17,18], $\sigma_Y$ of the experimental steel is attributed to multiple strengthening mechanisms, which can be expressed using the following formula:

$$\sigma_Y = \sigma_0 + \sigma_d + \sigma_g + \sigma_s + \sigma_p \tag{4}$$

where $\sigma_0$ is the internal frictional stress of body centered cubic (BCC) iron 54 MPa [19], $\sigma_d$ is the dislocation strengthening in martensite; $\sigma_s$ is the solid solution strengthening, $\sigma_g$

is the effective grain boundary strengthening, and $\sigma_p$ is the precipitation strengthening contribution from nano-scale precipitates.

The dislocation density can be estimated by the modified Williamson–Hall (MWH) method based on an XRD analysis [20,21]. The MWH equation is written as follows:

$$\frac{2\delta \cos\theta}{\lambda} = \frac{0.9}{D} + Mb\sqrt{\frac{\pi\rho}{2}}\frac{2\sin\theta}{\lambda}C^{1/2} \tag{5}$$

where, $\delta$, $\theta$, and $\lambda$ represent the XRD full width at half maximum (FWHM) of the diffraction peak, the diffraction angle, and the wavelength of the X-ray, respectively. For Cu radiation, the value of $\lambda$ is 0.154 nm. $D$, $\rho$, and $b$ are the average grain size, dislocation density and the Burgers vector of 0.248 nm, respectively. $M$ is a constant of 3, and it depended on the effective out cut-off radius of dislocation density.$C$ is the contrast factor of the dislocations and it can be expressed as follows [22]:

$$C = 0.285\left[1 - q\frac{h^2k^2 + k^2l^2 + l^2h^2}{(h^2 + k^2 + l^2)^2}\right] \tag{6}$$

where $h$, $k$, and $l$ are the Miller's indices of each peak of martensite.

In the present work, the XRD diffraction peaks used for this estimation were the (110), (200) and (211) peaks, as shown in Figure 9a. Using Equations (5) and (6) combined, the value of $0.9/D$ was obtained as the intercept in the Figure 9b, imposing a linear relationship between $2\sin\theta/\lambda$ and $2\delta\cos\theta/\lambda$ by Origin data analysis software (OriginLab, Northamptom, MA, USA). The parameter $q$ can be derived from the ratio between absolute value of slope and the intercept in Figure 9c, according to the linear relationship between $(2\delta\cos\theta/\lambda - 0.9/D)^2/(2\sin\theta/\lambda)^2$ and $(h^2k^2 + k^2l^2 + l^2h^2)/(h^2 + k^2 + l^2)^2$. Then, based on the value of the slope, as shown in Figure 9d, the dislocation density was calculated as $1.51 \times 10^{14}$ and $2.32 \times 10^{14}$ m$^{-2}$ for steel A and steel B by the fitted curves of $2\sin\theta \cdot C^{1/2}/\lambda$ and $2\delta\cos\theta/\lambda$, as shown in Figure 9d.

The increased yield stress resulting from the dislocation strengthening can be estimated by the Baile–Hirsch relationship [23]:

$$\sigma_d = MG\alpha b\rho^{1/2} \tag{7}$$

where $G$ is the shear modulus of 82 GPa, and $\alpha$ and $\rho$ are constants with the value of about 0.24 and 3 [19]. $\rho$ is the total dislocation density. Based on Equation (7), the $\sigma_d$ of steel A and B was calculated as 179.9 MPa and 223.0 MPa.

The $\sigma_s$ contribution is expressed using the following empirical equation [24]:

$$\sigma_s = 4570Xc + 84XSi + 32XMn - 30XCr + 80XTi \tag{8}$$

where $Xc$, $XMn$, $XSi$, $XCr$, and $XTi$ are the weight percentages of C, Mn, Si, Cr, and Ti dissolved in the matrix, respectively. The average content of $Xc$, $XMn$, $XSi$, $XCr$, and $XTi$ were obtained via SEM and EDS analyses, which were performed using at least five-spot analyses per condition, as shown in Table 3. The carbon content is difficult to accurately detect by EDS. It can be seen from Equation (8) that a small amount of carbon would make a great contribution to the solution strengthening. Therefore, the solution strengthening effect evaluated would be lower than the actual contribution value. The calculated $\sigma_s$ is shown in Table 3.

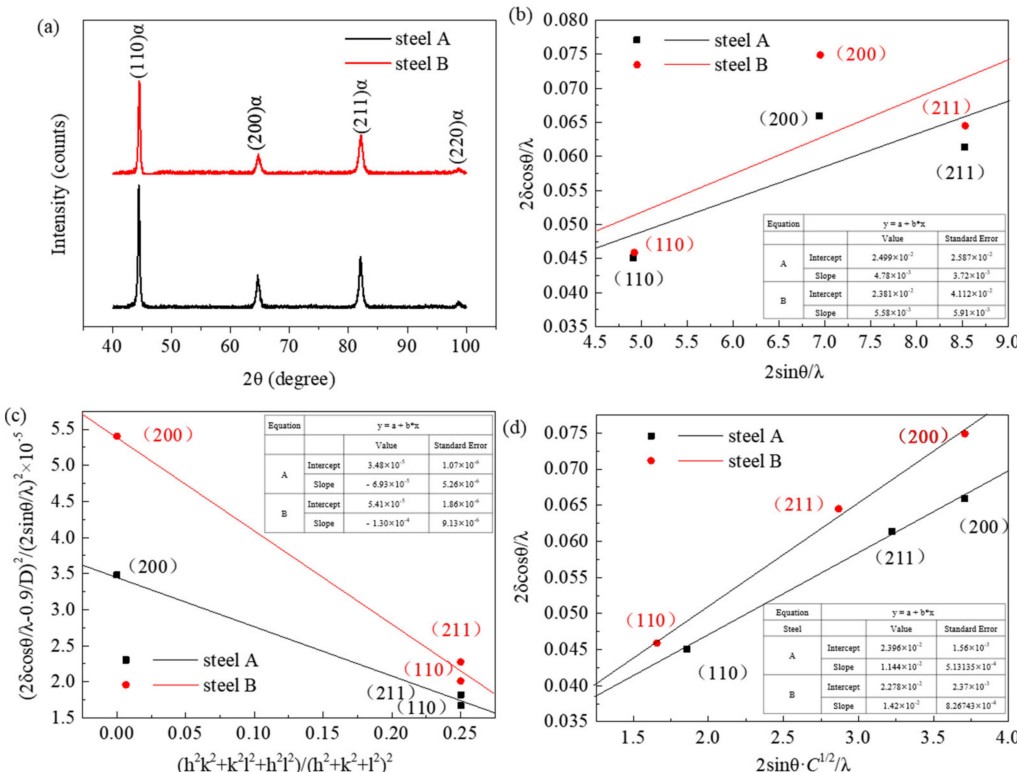

**Figure 9.** (**a**) The measured XRD profiles, and the estimation of dislocation density for steel A and B heated to 880 °C then air cooling according to a linear relationship between $2\sin\theta/\lambda$ and $2\delta\cos\theta/\lambda$ (**b**), $(h^2k^2 + k^2l^2 + h^2l^2)/(h^2 + k^2 + l^2)^2$ and $(2\delta\cos\theta/\lambda - 0.9/D)^2/(2\sin\theta/\lambda)^2$ (**c**), $2\sin\theta \cdot C^{1/2}/\lambda$ and $2\delta\cos\theta/\lambda$ (**d**).

**Table 3.** Chemical composition of the matrix (wt%) of steel A, B and the calculated $\sigma_s$.

| Samples | Mn | Si | Cr | Ti | $\sigma_s$/MPa |
|---------|------|------|------|-------|-------|
| A | 2.30 | 1.32 | 1.03 | 0.010 | 154.4 |
| B | 2.36 | 1.45 | 1.04 | 0.013 | 167.2 |

The $\sigma_g$ is calculated using the Hall–Petch principle [25]:

$$\sigma_g = k \times d^{-1/2} \tag{9}$$

where $k$ is the Hall–Petch slope 120 MPa/$\cdot\mu m^{1/2}$, and $d$ is the average width of the martensite lath in μm. According to the analysis of the microstructure in Figure 4, the average widths of the lath martensite of steel A and B are 0.367 μm and 0.332 μm, respectively. Therefore, the calculated results of $\sigma_g$ are 198.1 and 208.3 MPa for steel A and B, respectively.

The precipitation strengthening caused by carbide can be calculated quantitatively by the Ashby–Orowan equation [26] under the assumption of particle by-passing, as follows:

$$\sigma_p = \left(\frac{0.538Gb\sqrt{V_f}}{X}\right)\ln\left(\frac{X}{2b}\right) \tag{10}$$

where $X$ and $V_f$ are the mean diameter of the precipitates and the volume fraction of the precipitates, respectively.

Combined with the particle size observed in the TEM analysis, as shown in Figure 5, and the volume fraction of these particles calculated by Thermo-Calc software with the TCFE10 database, the resulting strength increase was calculated using Equation (10) and is shown in Table 4.

**Table 4.** Calculated volume fraction, average size of the carbide, and the calculated $\sigma_p$ of the experimental steel.

| Samples | Volume Fraction | Average Size/nm | $\sigma_p$/MPa |
|---------|-----------------|-----------------|----------------|
| A | $1.937 \times 10^{-4}$ | 9.676 | 47 |
| B | $2.376 \times 10^{-4}$ | 10.231 | 50 |

Based on the above calculations, the $\sigma_Y$ estimated by Equation (4) is about 633.4 and 702.5 MPa for steel A and B. It is far below the measured yield strength of 843 and 916 MPa.

According to Equation (8), the $\sigma_s$ will obviously increase if minor carbon is considered. So, the solid solution strengthening is the dominating strengthening mechanism for the experimental steels, and steel B with the higher carbon content in the matrix shows a higher yield strength.

### 3.2. Effects of Different Heating Temperaturse on Microstructures and Mechanical Properties

As shown in Figure 10, when the heating temperature decreases from 880 °C to 800 °C, the size of the martensite lath becomes shorter due to the refinement of the grain, and the amount of well-tempered martensite decreases. In addition, the recrystallization of ferrite was found in the samples that were soaked at 200 °C for 3 min then air-cooled. According to Table 5, the PSE of the steels under air cooling is better than that under water quenching, the value of which is similar to that obtained for the steels heated at 880 °C.

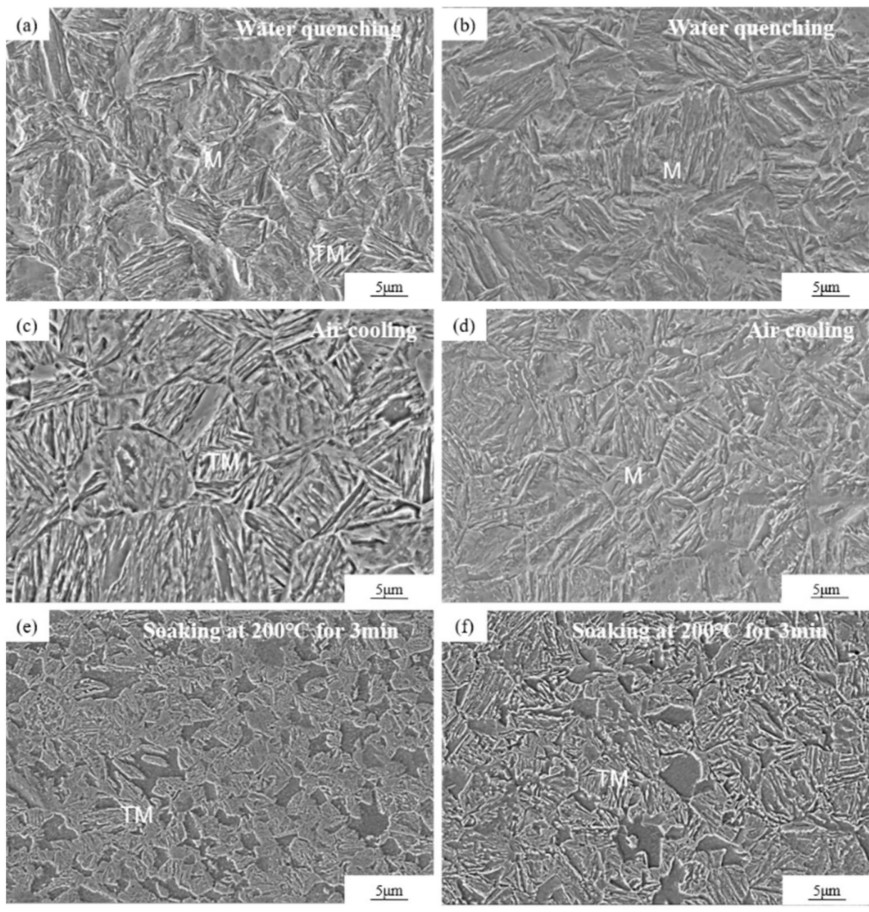

**Figure 10.** Microstructures of steel A (**a,c,e**) and steel B (**b,d,f**) heated to 800 °C for 3 min then treated by different cooling processes.

**Table 5.** Mechanical properties of steels heated to 800 °C for 3 min then under different cooling processes.

| Cooling Process | TS (MPa) | | YS (MPa) | | TE (%) | | PSE (MPa·%) | |
|---|---|---|---|---|---|---|---|---|
| | A | B | A | B | A | B | A | B |
| Water quenching | 1491 ± 22 | 1726 ± 43 | 1069 ± 33 | 1225 ± 19 | 7.6 ± 1.1 | 6.9 ± 0.3 | 11,332 | 11,909 |
| Air cooling | 1271 ± 18 | 1553 ± 11 | 828 ± 42 | 952 ± 17 | 9.0 ± 0.4 | 8.2 ± 0.2 | 11,439 | 12,735 |
| Soaking at 200 °C for 3 min | 1181 ± 27 | 1517 ± 15 | 640 ± 23 | 1056 ± 28 | 15.0 ± 0.7 | 10.2 ± 0.5 | 17,715 | 15,473 |

The equilibrium composition of austenite in steel A and steel B at 800 °C was calculated by Thermo-Calc software (Thermo-Calc Software, Stockholm, Sweden) with the TCFE10 database and is shown in Table 6, where it can be seen that the carbon content in austenite in steel A and B is 0.179 wt% and 0.210 wt%, respectively., and both values are higher than those in the matrix. The $M_S$ of steel A and B can be calculated as 396 °C and 369 °C, respectively by MUCG83 [27], which are close to the experimental results listed in Table 1. However, according to the calculated composition of austenite listed in Table 6, the $M_S$ of steel A and B heated at 800 °C can be calculated as 353 °C and 344 °C, respectively, so smaller amounts of auto tempered martensite were obtained due to the decrease in the $M_S$. It is obvious that the PSE of steels heated at 880 °C under air cooling is much higher than those at 800 °C. The PSE of steel A heated at 800 °C under air cooling is 11,439 MPa·%, a bit higher than that under water quenching, which is 11,332 MPa·%. However, the PSE of steel A after soaking at 200 °C for 3 min is 17,715 MPa·%, much higher than that after water quenching; this is because the interstitial carbon atoms can diffuse more easily from the martensite lattice during soaking. which leads to an increased elongation through the auto-tempering of martensite.

**Table 6.** Calculated equilibrium composition of austenite and the volume fraction of ferrite and austenite in experimental steels at 800 °C.

| Steel | Ferrite (%) | Austenite (%) | Elements in Austenite (wt%) | | | |
|---|---|---|---|---|---|---|
| | | | C | Mn | Si | Cr |
| Steel A | 30.3 | 69.7 | 0.179 | 2.47 | 1.29 | 1.04 |
| Steel B | 16.2 | 83.8 | 0.210 | 2.31 | 1.36 | 1.03 |

In order to further investigate the effect of the auto-tempering behavior of martensite on the properties of experimental steels heated at different temperatures, the hole expansion ratio of steels is listed in Table 7 and the crack shapes in the hole-edge regions of steels after HET are shown in Figure 11. The main crack around the hole-edge region after HET seems to occur along RD. The HER is 25.9% and 16.6% in steel A, which are obviously higher values than those for steel B. Moreover, the HER of the steels heated at 880 °C is better than that at 800 °C.

As was mentioned above, compared with steel B, the higher martensitic transformation temperature of steel A promotes the auto-tempering of martensite, and the well-tempered martensite can be more severely deformed compared to the less-tempered martensite. The HER of the steel heated at 800 °C is lower, which can be mainly attributed to the hardening of the martensite that could accelerate crack initiation at the interface of martensite and ferrite because of the large difference in hardness between the two phases [28]. The HER of the steel A heated at 880 °C is much higher because of the softening of its well-tempered martensite.

**Table 7.** HER of steel A and B under different temperatures.

| Heating Temperature | Hole Expansion Ratio (%) | |
|---|---|---|
| | Steel A | Steel B |
| 800 °C | 16.6 | 2.4 |
| 880 °C | 25.9 | 8.9 |

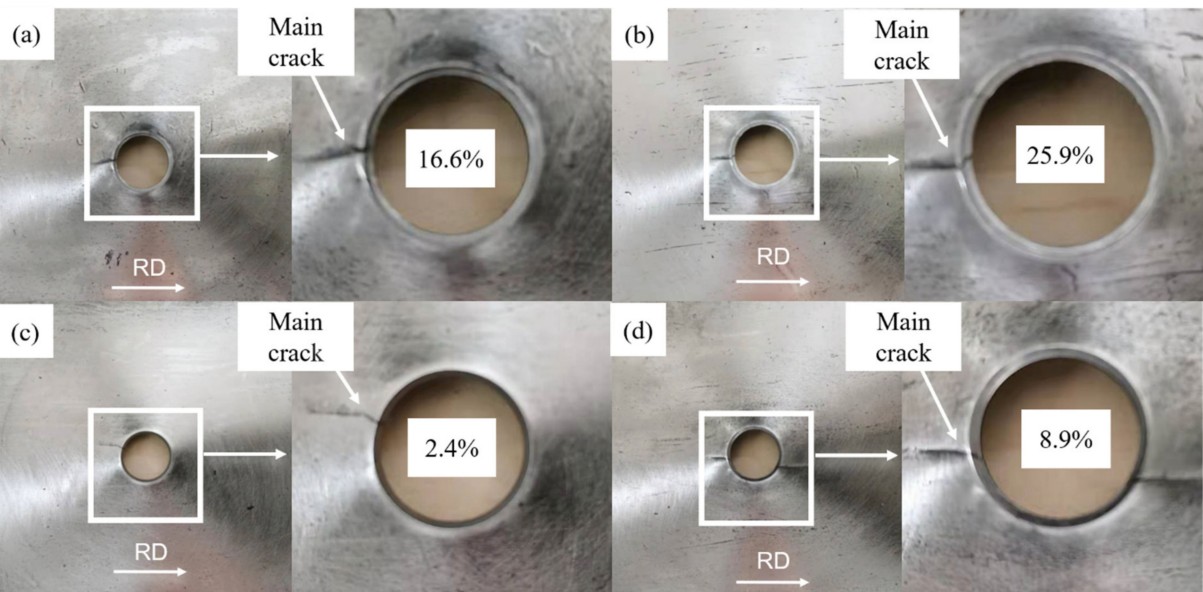

**Figure 11.** Crack shape in the hole-edge regions after HET of steels A (**a**,**b**) and steel B (**c**,**d**), heated at 800 °C and 880 °C for 3 min then air cooled.

### 3.3. Effects of Flash Heating on Microstructures and Mechanical Properties

Recently, flash heating technology with a heating rate of 100–300 °C/s has been proposed to study the effects of the mechanical properties of AHSS [29]. Flash heating is found to effectively refine the multiphase microstructures of DP steels and QP steels resulting from a retardation of recrystallization to a large extent and the induction of explosive nucleation of intercritical austenite [30–32].

The influence of conventional heating and flash heating on the structure and mechanical properties of the experimental steels is compared. The microstructures of steel A and steel B composed of martensite and ferrite are shown in Figure 12. Compared with the steels under conventional heating conditions, the refined martensite microstructure with equiaxed ferrite of the steels under flash heating was obtained and there was no obvious auto-tempering phenomenon because flash heating can result in the transformation of ferrite to austenite to be delayed and can raise the transit temperature above $A_3$ temperature; the $M_S$ of austenite is decreased accordingly [33]. As shown in Figure 13, the size of the martensite lath of steel A and B under flash heating was about 0.349 μm and 0.273 μm, which are also smaller sizes than those obtained under conventional heating. The prior austenite grain morphology and the size distribution of steels under different heating conditions are shown in Figures 14 and 15, where it can be seen that the the average size of the austenite grain was 12.19 μm and 12.83 μm under conventional heating, and 6.28 and 6.39 μm under flash heating. According to Equation (9), the effect of the grain boundary strengthening can be calculated as 203.1 MPa and 229.6 MPa.

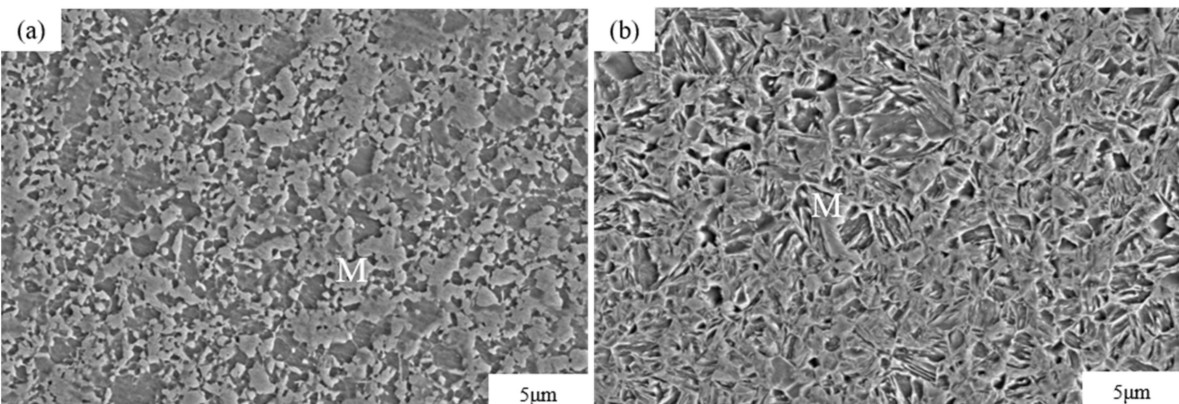

**Figure 12.** Microstructures of steel A (**a**) and B (**b**) under flash heating to 880 °C then air cooled.

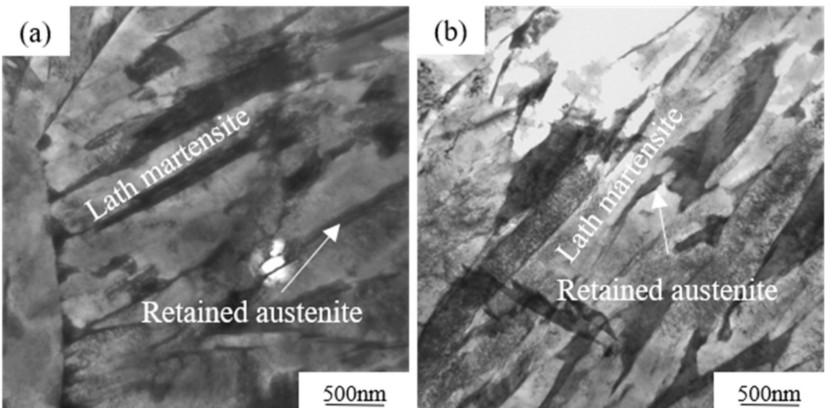

**Figure 13.** TEM analysis of lath martensite in steel A (**a**) and steel B (**b**) under flash heating to 880 °C then air cooled.

As shown in Figure 16b, the yield strength of the steels that were flash heated is obviously improved compared with that under conventional heating and air cooling. According to the XRD analysis in Figure 17a, the value of $0.9/D$ could be obtained as the intercept in the Figure 17b, the parameter $q$ can be derived from the ratio between the absolute value of slope and the intercept in Figure 17c, and based on the value of the slope, the dislocation density can be calculated as $1.80 \times 10^{14}$ m$^{-2}$ and $2.65 \times 10^{14}$ m$^{-2}$ for steel A and steel B, as shown in Figure 17d. Then the increased yield stress that resulted from the dislocation strengthening can be calculated by Equation (7). The values of $\sigma_d$ of steel A and B under flash heating are 196.4 MPa and 238.4 MPa, respectively. It can be seen that under flash heating, the effect of the grain boundary strengthening, and dislocation strengthening is not obvious for the experimental steel, while the yielding strength of the sample is about 100 MPa higher than that under conventional heating. As was stated in the above discussion, under flash heating, the carbon content in the martensite of steel is higher because of the existence of ferrite, so the contribution of solid solution strengthening is the main reason for the higher yield strength of the steels.

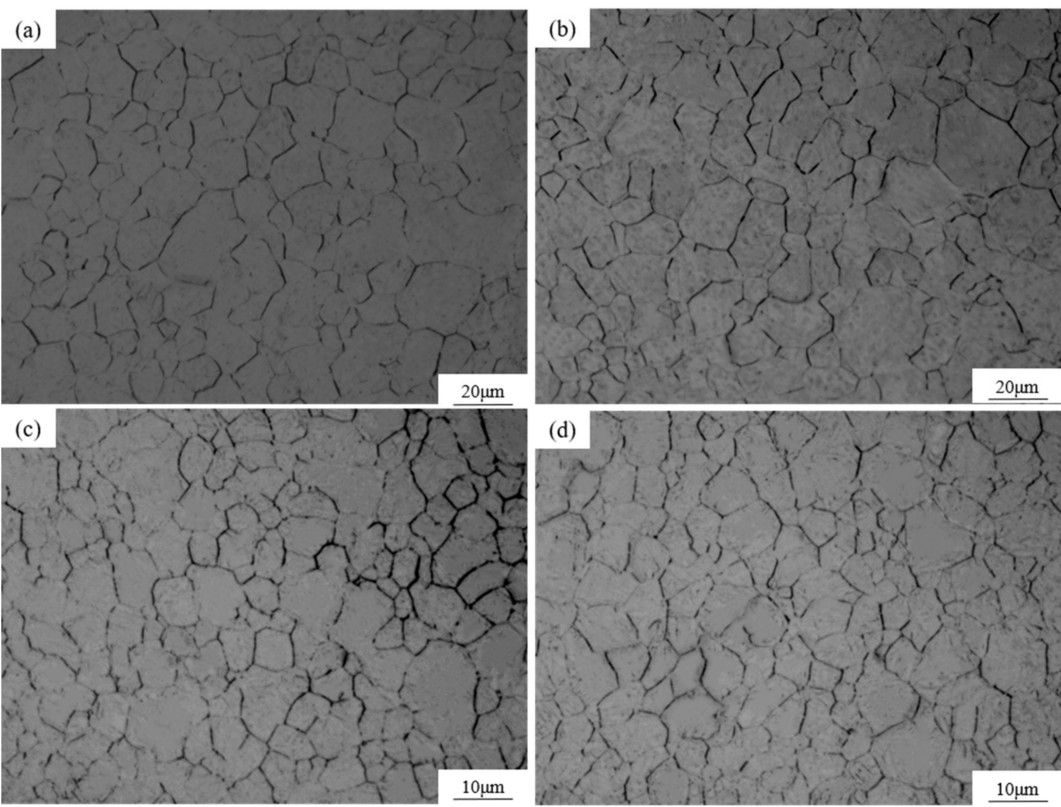

**Figure 14.** Prior austenite grain morphology of steels observed by OM: (**a**) steel A and (**b**) steel B under conventional heating at 880 °C then air cooled (CHA); (**c**) steel A and (**d**) steel B under flash heating to 880 °C then air cooled (FHA).

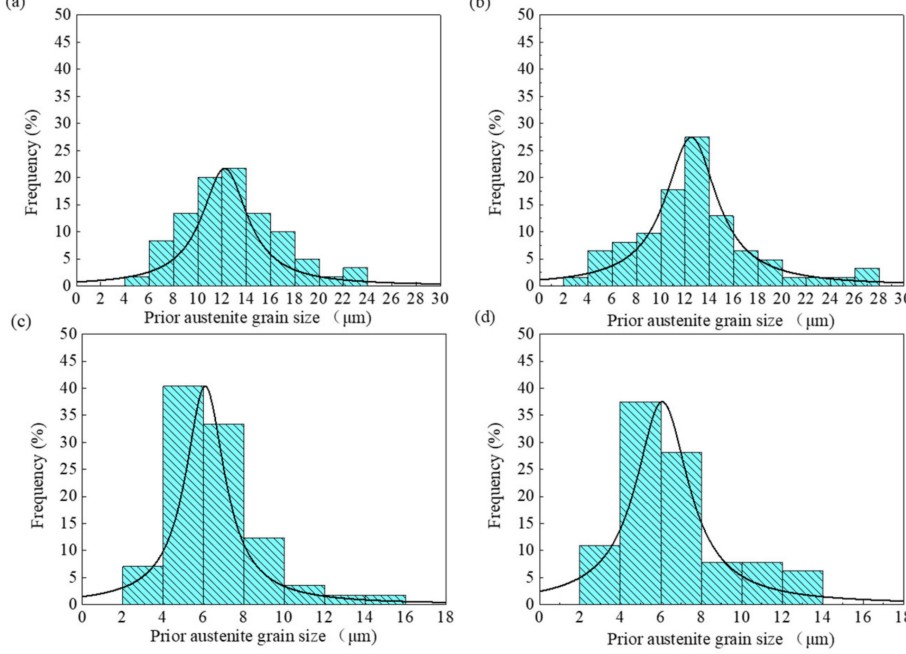

**Figure 15.** Prior Austenite grain size distribution of steels: (**a**) steel A and (**b**) steel B under conventional heating at 880 °C then air cooling (CHA); (**c**) steel A and (**d**) steel B under flash heating to 880 °C then air cooling (FHA).

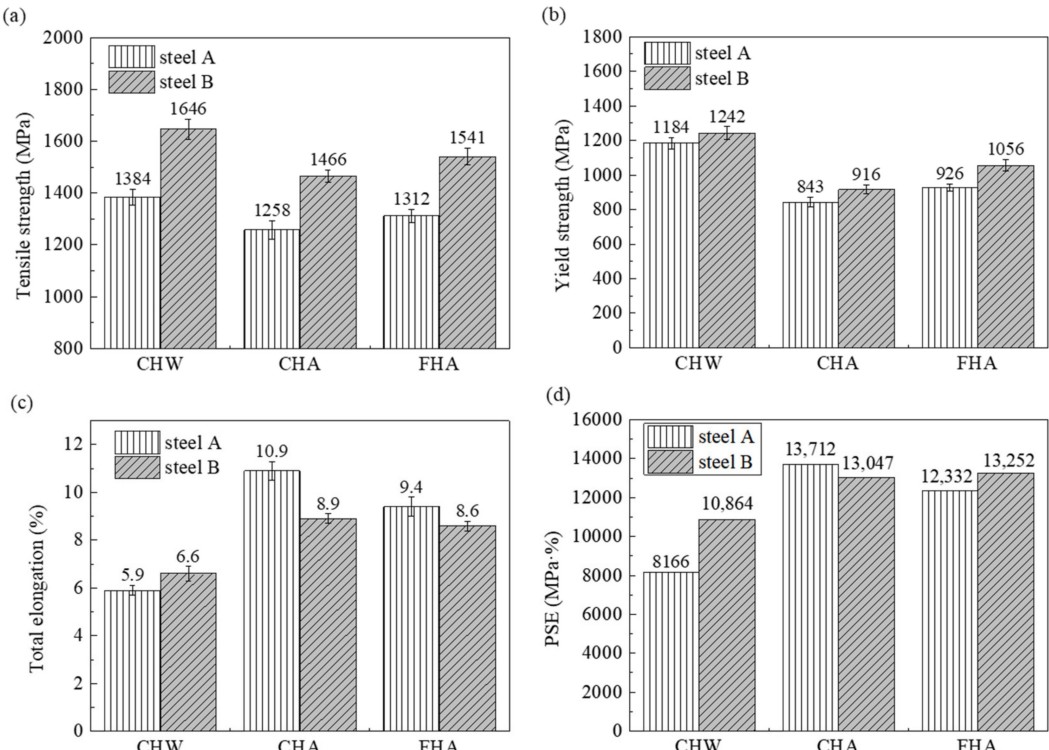

**Figure 16.** Mechanical properties of steels: (**a**) tensible strength, (**b**) yield strength, (**c**) total elongation and (**d**) PSE under conventional heating at 880 °C then water quenching (CHW), air cooling (CHA) and flash heating to 880 °C then air cooling (FHA).

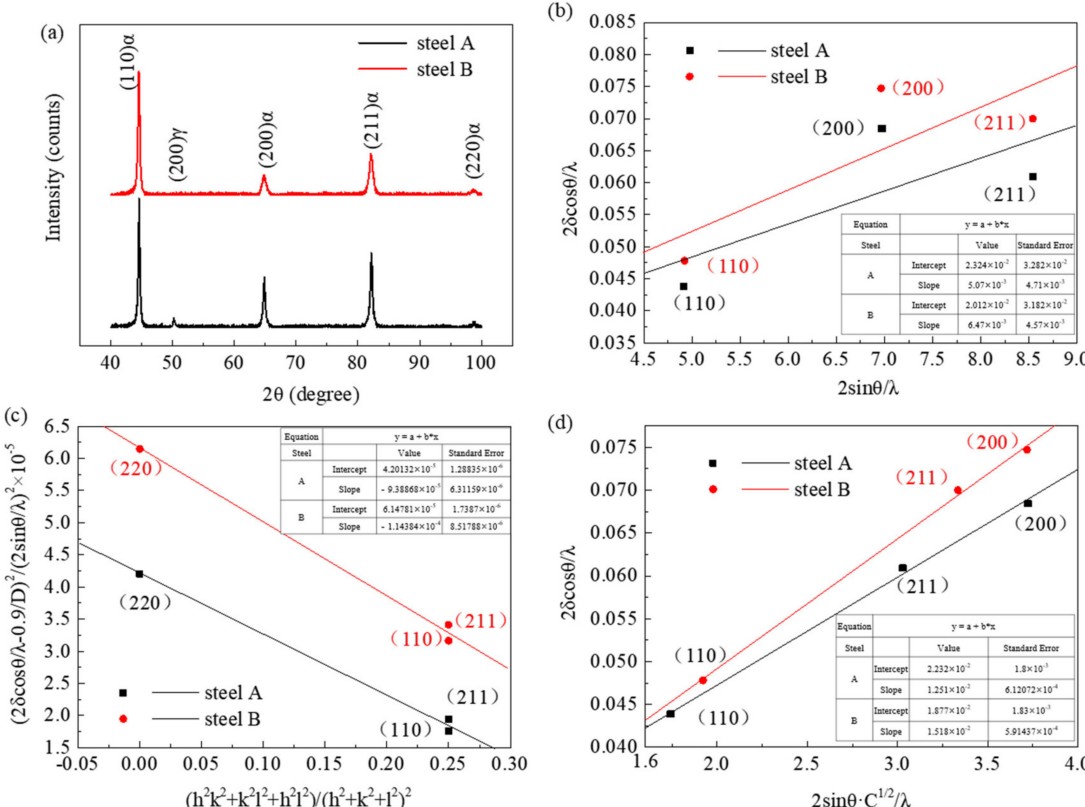

**Figure 17.** (**a**) The measured XRD profiles, the estimation of dislocation density for steel A and B under flash heating to 880 °C then air cooling according to a linear relationship between $2\sin\theta/\lambda$ and $2\delta\cos\theta/\lambda$ (**b**), $(h^2k^2+k^2l^2+h^2l^2)/(h^2+k^2+l^2)^2$ and $(2\delta\cos\theta/\lambda-0.9/D)^2/(2\sin\theta/\lambda)^2$ (**c**), $2\sin\theta\cdot C^{1/2}/\lambda$ and $2\delta\cos\theta/\lambda$ (**d**).

As is well known, the elongation of steel could be improved through uniform plastic deformation in a refined matrix. However, for the steels under flash heating, the elongation of steel A and B was lower than that under conventional heating as shown in Figure 16c. The grain refinement of the steel under flashing heating would enhance the thermal stability of austenite [34]. Moreover, although the heating temperature was 880 °C, far higher than its $A_3$ temperature, the ferrite phase still remained in the microstructure of the steel, which increased the carbon content in austenite. With the stability of the undercooling austenite increased, the $M_S$ of the experimental steel is so low that the martensite transformation is suppressed and hard to auto-temper. So, the retained austenite can be obviously observed in the microstructure of steel B, as shown in Figure 13. It is reported that Q&P steel under flash heating exhibited good elongation resulting from the increase in the content and stability of the retained austenite [34]. However, the elongation of the experimental steel was not improved. The relationship between the instantaneous strain hardening exponent (*n* value) and the true strain of the two steels was obtained according to the engineering stress–strain curve, as shown in Figure 18. It can be seen that the n value decreases constantly. According to the authors of [35], if transformation-induced plasticity happened in experimental steel, there would be a platform on the n value–true strain curve, resulting from transformation hardening and stress relaxation softening that coexist in the matrix during the gradual transformation of retained austenite to martensite. Obviously, there is no transformation-induced plasticity effect for the retained austenite in the steels during the tensile deformation. So, although the grain refinement can be found in the samples under flash heating, the microstructure composed of the ferrite soft phase and the less tempered martensite hard phase easily cracked under tensile stress. On the contrary, the samples under conventional heating had a better elongation because of their microstructure which is composed of well-tempered martensite.

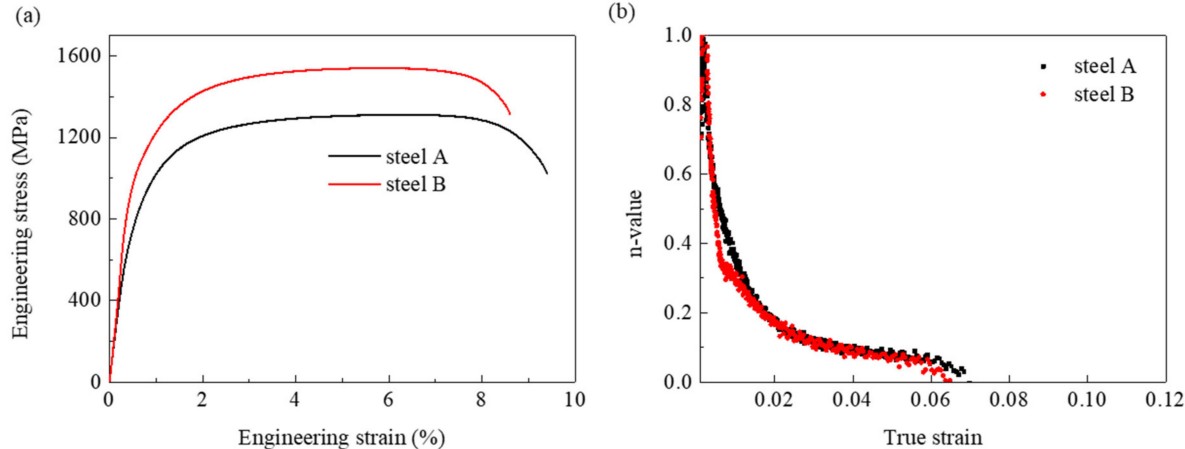

**Figure 18.** (**a**) Engineering stress versus strain and (**b**) *n*-value versus true strain of steel A and B under flash heating.

## 4. Conclusions

(1) Different cooling processes have a great influence on the performance of the investigated steels resulting from the different auto-tempering effect. Compared with the steels under water quenching, there was approximately a 70% increase of the PSE of steel A with well-tempered martensite under air cooling, which was caused by the fact that the probability of auto-tempering was less for the steel under water quenching according to the calculated average diffusion distance of the carbon atom in martensite for each steel. For steel B with a higher carbon content, its PSE was only 20% higher under air cooling than that of water quenching because its $M_S$ is lower and this limits the auto-tempering of martensite. A tensile strength of over 1400 MPa and a total elongation rate exceeding 10% can be obtained for steel B under soaking at 200 °C for 3 min.

(2)  Steels heated at 800 °C then treated by either air cooling or water quenching both exhibited a low PSE since the $M_S$ temperatures of the austenite in these steels were low. On the contrary, the PSE of the steels can be obviously improved under soaking at 200 °C for 3 min where the well-tempered martensite dominated. Although the steel with the microstructure composed of the ferrite, well-tempered martensite, and less-tempered martensite had an elongation that exceeded 15%, the hole expansion ratio was still lower because of the undesirable hardness distribution between the phases.

(3)  Flash heating obviously refines the microstructure with equiaxed ferrite. The contribution of solution strengthening is the main cause of the higher yield strength for the steels under flash heating. Moreover, the improvement of elongation could not be obtained because cracks easily occurred due to the stress concentration at the interface of the soft phase ferrite and the hard phase martensite. It was difficult to obtain steel with ideal properties and less temped martensite under flash heating.

**Author Contributions:** Conceptualization, Y.H. and H.J.; methodology, Y.H.; software, H.J.; validation, H.J.; formal analysis, L.L. (Li Lin); investigation, R.L.; resources, Y.H.; data curation, W.Z.; writing—original draft preparation, H.J.; writing—review and editing, H.J.; visualization, Y.Z.; supervision, L.L. (Lin Li); project administration, Y.H.; funding acquisition, Y.H. All authors have read and agreed to the published version of the manuscript.

**Funding:** The present work was supported by the National Key R&D Program of China (Grant No. 2017YFB0304402) and National Natural Science Foundation of China (Grant No. 51971127).

**Informed Consent Statement:** Informed consent was obtained from all subjects involved in the study.

**Conflicts of Interest:** The authors declare no conflict of interest.

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
