# Peer review of "Microstructures and Properties of Auto-Tempering Ultra-High Strength Automotive Steel under Different Thermal-Processing Conditions"

_metals, doi:10.3390/met11071121_

Round 1

Reviewer 1 Report

Dear authors, thanks for sharing your work with the rest of the community. 

The work is interesting and fits very well with worldwide research interest in simultaneous weight and cost reduction. It is very enriching in terms of a proposing the mechanisms behind the experimental observations. Nevertheless some extra information for reproducibility by third party researchers and framing the final use would help better understanding the reach of the results. Please find below my comments I this sense:

Please include final condition use information. Are the steels intended to be quenched in their final shape, hot pressed or cold formed in the HT condition? This info is important to understand the aim of the work. 

There is some relevant information missing information on the materials: S, P, Al contents, employed furnace vacuum level and achieved N and H values, hot forging/rolling reduction, cold rolling reduction program and annealing time+temperature and atmosphere if used, skin passing and departure mechanical  properties prior to HT. 

Chemical composition and transformation temperature uncertainties are missing. 

Cooling rate information is missing: air/water temperature, specimen position when soaking, agitation, hot soaking media definition and temperature uncertainty.

As HET standard is mentioned, tensile testing and grain size measument standards should also be mentioned. Strain rate should be mentioned instead of cross head speed and gage length. Specimen width is missing and required to asses if the gage length is appropriate.

200°C soaking process unclear: direct quenching in a media at 200 for 3 min/30 min or air cool to 200 and then soaking of 3/30 min?

Missing information in tensile test results: number of repeats, measurement uncertainty, Ag or Agt, maybe even n. PSE use for toughness is arguable, the term energy absorption should be used instead. Effective energy to fracture would have been a better choice to asses toughness. 

Why was taken BBC friction for sigma zero provided tempered marsite is tetragonal, martensite monoclinic?

I look forward to have my comments answered.

Regards

Author Response

Dear sir or madam 

Thank you for giving us  your positive and constructive comments and suggestions on our manuscript entitled “Microstructure and properties of auto-tempering ultra-high strength automotive steel under different thermo-processing"We have tried our best to revise our manuscript according to the comments.

Reviewer 2 Report

Here a study of the autotempering behavior of high strength steels is investigated. Two alloys are selected to vary Ms and multiple heating/cooling profiles are used to vary the propensity for autotempering. The resulting mechanical properties are assessed in terms of the microstructure, various strengthening mechanisms, and the effects of autotempering.

Overall, the observed control over the mechanical properties via autotempering is of value. However, the manuscript suffers from poor writing and several technical deficiencies. Specific comments regarding these issues are included below.

  • Many grammatical issues are found throughout. The manuscript must be carefully checked to ensure grammatical correctness and to improve the readability. For instance on line 72, one does not conduct a “MTS C45.305E.” Rather a tensile test is conducted using a MTS C34.305E.
  • The organization of the manuscript should be changed to improve readability. The methods section and experimental results should be changed to agree better in terms of the order of presentation. In addition, it is recommended to move dilatometer results to earlier in the methods as this is used to determine the various temperatures used in the heating/cooling treatments.
  • From Figure 1, it appears that Ac3 was determined during cooling as opposed to heating. This must be clarified by either altering the figure to show Ac3 being determined on heating or changing to a measurement of Ar3.
  • For Figure 2, 4, 8, 10, and 11, please indicate which micrographs are OM and which are SEM to the reader in interpreting the contrast. In addition, additional labels of features within the micrographs (i.e. regions of retained austenite, martensite, tempered martensite) would be helpful as these are often cited in the text.
  • The diffusion analysis in Figure 6 requires significant clarification. How is the time determined for the calculation? Are the cooling rates known for the different cooling treatements? Why does the diffusion distance decrease with decreasing temperature? The text states that the figure should show the accumulated diffusion distance, which necessarily increases with time (and hence additional cooling). It appears that Figure 6 may simply show the average diffusion distance at a given temperature.
  • The dislocation density measurement is appears to have a number of technical issues. First, has the instrumental broadening been accounted for? This may introduce significant error into the measurement. Second, the Williamson-Hall method is well known to be flawed (See The effect of dislocation contrast on x‐ray line broadening: A new approach to line profile analysis, Applied Physics Letters 69 (21) (1996) 3173-3175.). Figure 7b and 14b are clearly not well fit by the Williamson-Hall method and so a different approach is required. Finally, the domain size, D, is incorrectly used. D is not related to the carbides rather it is the martensite/ferrite grain size. Hence it is inappropriate to fix the domain size (D) at 0. The martensitic microstructure has features at multiple length scales (martensite lathe, martensite packets, parent grains, etc.) that will disrupt coherent scattering thereby introducing some finite domain size. In summation this calls into question the relative impact of the dislocation density as compared to other strengthening mechanisms.
  • Similarly for the discussion of the Hall-Petch strengthening it is likely that the wrong grain size has been used. Recent analysis of the Hall-Petch effect in a variety of martensitic steels indicates that the martensite lathe or block size are what contribute to Hall-Petch strengthening (see The Effect of Lath Martensite Microstructures on the Strength of Medium-Carbon Low-Alloy Steel, Crystals 10 (3) (2020) 232.). This results in a underestimate of the grain size effect on strength.
  • Collectively, a proper analysis of the strengthening mechanisms (dislocation density, Hall-Petch, precipitates, etc) is absent. As a result it is difficult to assess the role of autotempering on the improved properties as the mechanisms cannot be identified from the current analysis.

Round 2

Reviewer 1 Report

Dear authors,

Thanks for your improvements in the document.

Would it be please possible a futher effort from your side to increase the significance of the results by introducing the uncertainties and/or standard deviations of the numerical results when applicable? Please revise too the number of significant digits used for reporting the experimental values (e.g. the UTS and YS values expressed).

Regards,

Author Response

Dear Sir or Madam:

Thanks so much for your suggestion. The uncertainties of the numerical results should be introduced and listed in table2 and table5. The number of significant digits used for reporting the experimental values should also be standard. The modification has been made and please see the revised paper in detail.

Best wishes

Reviewer 2 Report

The manuscript is much improved following revisions. A few minor modifications are needed.

-Please double check for typos

-Figure 3, the M and TM labels show very poor contrast with the micrograph and should be changed to improve visibility.

-The captions of Figure 9 & 17 need to be improved to actually describe the figure. Specifically, a better description for parts b-d is needed.

Author Response

Dear Sir or Madam:

Thanks so much for your suggestion.

Deeply sorry for the typos. The modification has been made and please see the revised paper in detail.

The M and TM label have been modified in figure 3. Please see the revised paper in detail.

The supplement of description for Fig9(b,c,d) and Fig17(b,c,d) has been made and please see the revised paper in detail.

Best wishes
